

# Up-regulation of cryptochrome 1 gene expression in cotton bollworm (*Helicoverpa armigera*) during migration over the Bohai Sea

Liyu Yang[1],[*], Yingjie Liu[1],[*], Philip Donkersley[2] and Pengjun Xu[1]

[1] Chinese Academy of Agricultural Sciences, Tobacco Research Institute, Qingdao, Shandong, China
[2] Lancaster Environment Centre, Lancaster University, Lancaster, UK
[*] These authors contributed equally to this work.

## ABSTRACT

Cryptochromes (CRYs) are flavoproteins and play a pivotal role in circadian clocks which mediate behavior of organisms such as feeding, mating and migrating navigation. Herein, we identified novel transcripts in *Helicoverpa armigera* of six isoforms of *cry1* and seven isoforms of *cry2* by Sanger sequencing. Phylogenetic analysis showed that the transcripts of *cry1* and *cry2* align closely with other insect *crys*, indicating within-species divergence of *Hacry*. A *dn/ds* analysis revealed that the encoding sequence of the *cry1* was under purifying selection by a strong negative selection pressure whereas the *cry2* was less constraint and showed a less strong purification selection than *cry1*. In general, *Hacrys* were more abundantly transcribed in wild migrating populations than that in laboratory maintained populations, and expression of the *cry2* was lower than *cry1* in all samples tested. Moreover, when compared with the migrating parental population, offspring reared in laboratory conditions showed a significant reduction on transcription of the *cry1* but not *cry2*. These results strongly suggest that *cry1* was more related to the migration behavior of *H. armigera* than *cry2*.

## INTRODUCTION

Cryptochromes (CRYs) genes were firstly characterized in *Arabidopsis thaliana* L. and then found in a wide range of organisms including animals and bacteria (*Ahmad & Cashmore, 1993*; *Lin & Todo, 2005*). According to their phylogenetic analysis, CRYs from different organisms can be divided into three subclasses: plant CRYs, animal CRYs and CRY-DASH (*Selby & Sancar, 2006*; *Lin & Todo, 2005*; *Daiyasu et al., 2004*). As receptors for blue and ultraviolet (UV-A) light, plant and animal CRYs mediate many light-dependent responses such as flowering in plants and circadian rhythms in animals (*Emery et al., 1998*; *Griffin, Staknis & Weitz, 1999*; *Guo et al., 1998*). Generally, CRYs have a high degree of sequence affinity to the members of DNA photolyase but do not possess any function on double-strand DNA repair, except for CRY-DASH, which is found

Corresponding author
Pengjun Xu, xupengjun@caas.cn

in many organisms except for insects and has a single-stranded DNA-specific photolyase activity (*Selby & Sancar, 2006*).

In insects, two forms of CRYs, Drosophila-type CRY (CRY1, acting as a photoreceptor) and mammalian-type CRY (CRY2, acting as a transcriptional repressor), represent three kinds of circadian clocks: type 1, containing only CRY1 as in Drosophila; type 2, containing only CRY2 as in species from Hymenoptera and Coleoptera and type 3, containing both CRY1 and CRY2 as in mosquito and species in Lepidoptera (*Zhu et al., 2005, 2008; Yuan et al., 2007; Wang et al., 2013; Chang et al., 2019*). Besides the regulation of circadian clocks, CRY proteins are also well-known for time-compensated sun compass navigation, such as in the monarch butterfly and *Drosophila* etc. (*Zhu et al., 2008; Merlin, Gegear & Reppert, 2009; Yoshii, Ahmad & Helfrich-Forster, 2009; Marley et al., 2014; Bazalova et al., 2016*). Differently to diurnal butterflies and *Drosophila*, most moths are nocturnal. *Cry* genes have been reported in several moth species (*Yuan et al., 2007; Chang et al., 2019*). The cotton bollworm, *Helicoverpa armigera* (Lepidoptera: Noctuide), is one of the most important long-distance migratory agricultural pests in China (*Wu & Guo, 2005; Feng et al., 2003, 2004, 2005, 2009*). Every year, the first generation of the cotton bollworm emerges in late June feeding on wheat in northern China (especially Shandong, Hebei, and Henan Provinces) and then the adults migrate to the northeast (Liaoning Province, China), where they reproduce one or two generations and the offspring migrates back to the southern China for overwintering (*Wu, Xu & Guo, 1998; Feng et al., 2005; Wu & Guo, 2005; Wu et al., 2001*). Recently, two types of CRYs were reported in *H. armigera*, named *Hacry1* and *Hacry2*, respectively (*Yan et al., 2013*).

Herein, we identified several novel isoforms of *cry* gene transcripts from *H. armigera*. We also investigated mutation and expression patterns of these genes with interests in: (1) evolutionary selection pressure acting on the two types of *cry* genes, and (2) comparison of transcription patterns between migrating and non-migrant cotton bollworms to obtain evidence whether CRYs are related to the migrating behavior in *H. armigera*.

## MATERIALS AND METHODS

### Insects and culture conditions

Migrating adult cotton bollworms (CD) were captured in different months in 2011 by a vertical pointing trap, on Beihuang Island in Bohai sea in Shandong province (38°23.200′N, 120°54.500′E). There were no local individuals of this species (*Feng et al., 2003, 2004, 2005*). Trapped insects were collected every 3 h starting at 18:00 pm (Beijing time) and were directly stored in liquid $N_2$. The population of LF (originally captured in Langfang, Hebei province, in 2005) was used as control and reared in laboratory at 25 °C with a light (L):dark (D) photoperiod of 14 h:10 h. These control insects were 3rd day post eclosion adults exposed to natural light cycles in July 2011. To investigate the influence of heredity on *Hacrys* expression, offspring of CD were reared in laboratory (population CDF3) and collected at 15:00 pm under natural light using 3rd day post eclosion adults in August 2011.

Additional three laboratory reared populations were used for obtaining *Hacrys* expression profiles in different population (Population-96s, captured in Xinxiang, Henan province, in 1996; Population-1463, captured in Xiajin, Shandong province, in 2004; Population-XXF1, captured in Xinxiang, Henan province, in 2011). Except for studying the developmental stages, all insects reared in lab were the 3rd day instar adults under natural light.

## Cloning of full length *Hacry* gene transcripts

Total RNA was isolated from *H. armigera* using TRIzol reagent (Invitrogen, Carlsbad, CA, USA); cDNA was synthesized using oligo(dT) and M-MLV Reverse Transcriptase (Promega, Madison, WL, USA). To identify transcripts of *cry* genes in *H. armigera*, degenerate primers based on insect *cry* genes (including *Hacry1* (GenBank no.: GQ896502) and *Hacry2* (GQ896503)) were used to amplify partial segments of *Hacry* gene transcripts by nested RT-PCR (Table S1), using the following PCR program: 30 s at 94 °C, 30 s at 55 °C, and 60 s at 72 °C for 40 cycles. To obtain complete transcript sequences of the *Hacry* genes, 5′ and 3′ RACE (Rapid Amplification of cDNA Ends) were performed using the SMART RACE cDNA Amplification Kit (Clontech, Palo Alto, CA, USA) with gene specific primers (Table S1).

To avoid possible artifacts, specific primers amplifying the complete ORFs (open reading frames) of *Hacrys* were designed according to the RACE results (Table S1). PCR program using the cDNA templates was: 30 s at 94 °C, 30 s at 55 °C, and 150 s at 72 °C for 40 cycles. PCR products were purified, cloned into the pEASY-T Cloning Vector (TransGen, Beijing, China). Positive clones of plasmids of *Hacry* were purified and sequenced by using Sanger sequencing.

## Sequence alignment and phylogenetic analyses

Percentage identities of the nucleotide and amino acid sequences were calculated using CLUSTAL W program (*Thompson, Higgins & Gibson, 1994*). The longest obtained sequences of *cry1* (*Hacry1-4*) and *cry2* (*Hacry2-6*) were selected to use together with the other insect *cry1s* and *cry2s* to determine selection pressures (Table S2).

The sequences were aligned and phylogenetically analyzed using maximum likelihood (ML) methodology implemented in RAxML 7.3.2 (*Stamatakis, 2006*) under the GTR + G substitution model. For estimating nodal support, nonparametric bootstrap proportions (*Felsenstein, 1985*) with 1,000 pseudoreplicates were used. For the phylogenetic inference, we divided the dataset into three partitions according to codon position in each gene (1st/2nd/3rd codon). The multiple partitions controlled for heterogeneity across dataset, such as variation in substitution rates. Three replicates were conducted for each analysis to assess whether runs failed to converge upon the optimal posterior distribution and whether likelihood values, branch lengths, tree topology or posterior probabilities differed among runs. We made two kind of trees. One contained only butterfly, moth and outgroup for selection analysis, and another one contained much more insect species for phylogenetic purpose.

## Branch and Branch-site test of selection

Selection can promote the evolution of sensory systems and make organisms to adapt to local ecological conditions. To investigate the potential signal of positive selection acting on the *cry* genes between nocturnal moths and diurnal butterflies, Maximum Likelihood approach (*Nielsen & Yang, 1998*) was employed to test for differences in selection pressure, using the CODEML program of PAML version 4.5 (*Yang, 1997*). Branch models and branch-site models were employed to detect positive selection on the lineages. These tests of selection were phylogeny-based tests with requirements of unrooted input-trees. The RAxML trees mentioned above were converted to unrooted trees. For branch models, five hypotheses were evaluated: (1) one *dn/ds* ratio for all branches; (2) *dn/ds* ratio = 1 for all branches; (3) moth lineage and butterfly lineage have different *dn/ds* ratio ($\omega_2$ and $\omega_1$); (4) neutral evolution for moth ($\omega_2 = 1$); and (5) the free-ratio model with free *dn/ds* ratio for each branch. For branch-site modes, the moth lineage was defined as foreground, and the rest lineages were defined as background branch, which were specified in the tree file by using branch labels. Likelihood ratio test (LRT) was used to investigate if the alternative model, indicating positive selection, was superior to the null model.

To investigate non-synonymous mutations within *Hacry1* and *Hacry2*, we calculated the average *dn/ds* ratio under one ratio model in PAML. In this analysis, all the detected isoforms of the *cry1* and *cry2* transcripts were used.

## Real-time PCR analysis

With β-*actin* (GenBank accession no.: X97614) as the reference gene, Real-time PCR (qPCR) reactions were carried out with the first strand cDNAs using the TaqMan method in 20 µl reaction agent comprised of one µl of template cDNA, 2*Premix Ex Taq$^{TM}$ (Takara, Kusatsu, Japan), 0.2 µM each primer and 0.4 µM probe (Table S1) on a 7500 Fast Real-time PCR System (Applied Biosystems, Foster City, CA, USA). Thermal cycling conditions were: 45 cycles of 95 °C for 15 s, 60 °C for 34 s. cDNA sample of each group was replicated three times. At least three groups of individuals were tested for each data-point. Fold differences of *Hacry1* or *Hacry2* transcripts were calculated according to the $2^{-\Delta\Delta CT}$ method (*Livak & Schmittgen, 2001*).

## Statistical analysis

Statistical analyses were conducted using the STATA package (version 9.0) and GraphPad InStat 3 (GraphPad Software, Inc., San Diego, CA, USA). The Student's *t*-test and/or ANOVA were used to determine the significant differences.

# RESULTS

## Transcripts of cry genes in *H. armigera*

By RT-PCR and RACE method, the complete sequences of *cry1* and *cry2* gene transcripts with variable 5′ and 3′ end sequences were obtained from *H. armigera* (Fig. 1). BLAST searches performed at NCBI revealed that these *Hacry* transcripts were highly similar to *cry1* or *cry2* from the other insects, respectively. The encoding nucleotide sequences of the
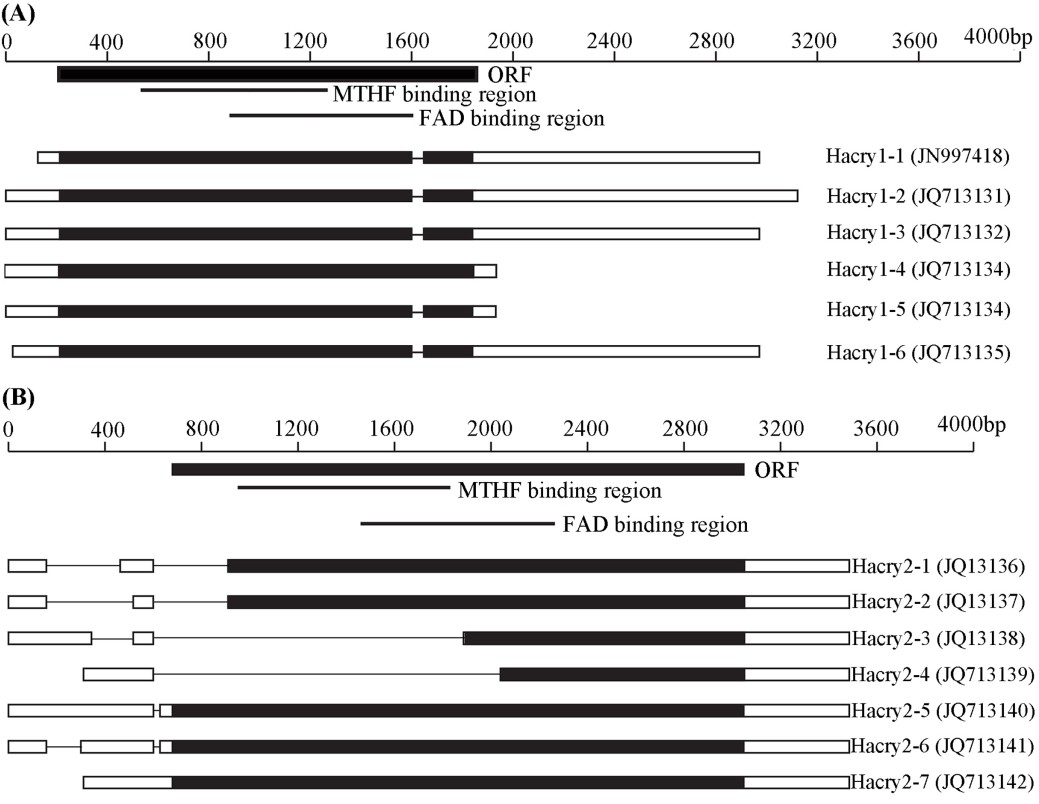

**Figure 1 Transcripts of *Hacry1* (A) and *Hacry2* (B) in *H. armigera*.** "▪" Stands for the open reading frame (ORF) region. "□" stands for untranslated region (UTR). "-" stands for gap. The conserved MTHF and FAD binding regions were shown on the top. The names and GenBank accession numbers of transcripts were shown on the right.

*cry1* of *H. armigera* displayed homology of 91–99.8% with each other whereas the *cry2* encoding sequences displayed much more polymorphisms including large deletions (Tables S3 and S4; Fig. 1; Fig. S1). All deduced amino acid sequences of the *cry1* contained fragments of the C-terminal extension (CCE) and the conserved photolyase homology region (PHR) which included a flavin adenine dinucleotide (FAD) domain and an additional pterin derivative domain (methenyltetrahydrofolate, MTHF) (Fig. 1; Fig. S1). However, *Hacry2-1*, *Hacry2-2*, *Hacry2-3* and *Hacry2-4* lacked a complete PHR encoding region (Fig. 1; Fig. S1).

## Phylogenetic analysis and selection pressure

Tree based analysis indicated that transcripts of *cry1* and *cry2* cloned in our study clustered with the one from species in Lepidoptera (Fig. 2). The free-ratio model provided a better fit for the two *cry* genes than the other four models. No significant differences in *dn/ds* were detected between the clades of moths and butterflies (Table 1). The branch-specific ω values of moths were 0.0287 and 0.0219 for *cry1* and *cry2*, respectively. The corresponding ω values of butterflies were 0.0409 and 0.0130, respectively (Fig. 2). Interestingly, for *cry1*, the *dn/ds* ratio of moths was lower than that of butterflies, suggesting increased constraint on nonsynonymous substitutions. In contrast, for *cry2*, the *dn/ds* ratio of moths

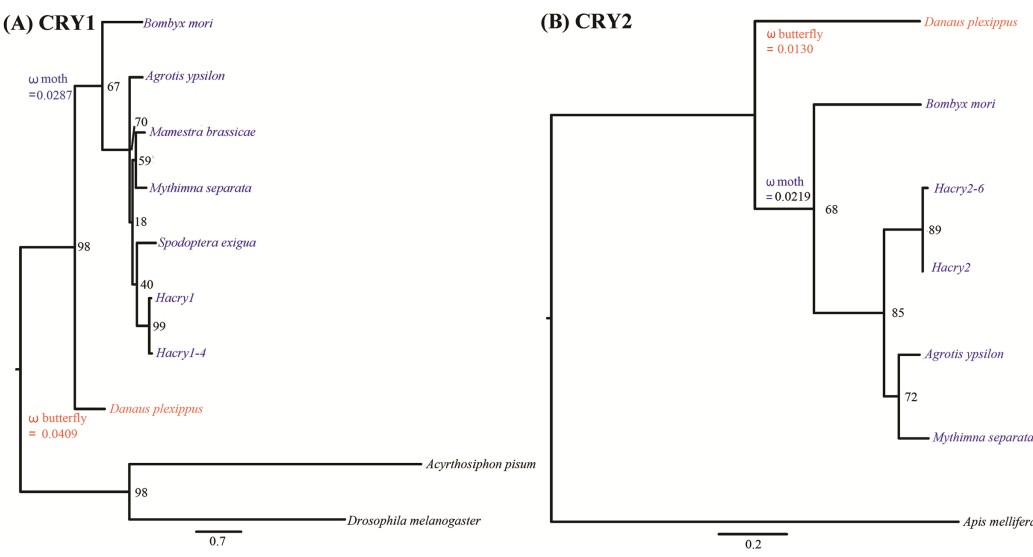

**Figure 2 Phylogenetic reconstruction of *cry* genes in species from Lepidoptera based on maximum likelihood.** (A) Tree based analysis of *cry1* genes using the ones of *Drosophila melanogaster* and *Acyrthosiphon pisum* as outgroup. (B) Tree based analysis of *cry2* genes using the one of *Apid mellifera* as outgroup. Values on the nodes are the nonparametric bootstrap proportions (MLBPs). Moths denoted in blue and butterflies in red. Branch-specific ω values are shown on nodes of the common ancestors. The names and GenBank accession numbers of *crys* were shown in Table S2.

**Table 1 Selective patterns for Cryptochrome genes.**

| Genes | Model | np[a] | Ln L[b] | Estimates of ω | Models compared | LRT[c] | *p*-values |
|---|---|---|---|---|---|---|---|
| Cry1 | **Branch model** | | | | | | |
| | A: one ratio | 19 | −9,129.37 | ω = 0.0278 | | | |
| | B: one ratio ω = 1 | 18 | −10,508.58 | ω = 1 | B vs. A | 2,758.42 | <0.001 |
| | C: the moth branch has $\omega_2$, the butterfly branch has $\omega_1$ | 21 | −9,127.01 | $\omega_2 = 0.0287$, $\omega_1 = 0.0409$ | A vs. C | 4.72 | 0.09 |
| | D: the moth branch has $\omega_2 = 1$ | 20 | −10,114.21 | $\omega_2 = 1$, $\omega_1 = 0.0270$ | D vs. C | 1,974.4 | <0.001 |
| | E: each branch has its own ω | 35 | −9,111.95 | Variable ω by branch | A vs. E | 34.84 | 0.004 |
| | **Branch-site models** | | | | | | |
| | G: the moth branch | 22 | −9,061.95 | | | | |
| | H: the moth branch has ω = 1 | 21 | −9,061.95 | | H vs. G | 0 | 1 |
| Cry2 | **Branch model** | | | | | | |
| | I: one ratio | 13 | −5,538.87 | ω = 0.0186 | | | |
| | J: one ratio ω = 1 | 12 | −6,629.16 | ω = 1 | J vs. I | 2,180.58 | <0.001 |
| | K: the moth branch has $\omega_2$, the butterfly branch has $\omega_1$ | 15 | −5,534.15 | $\omega_2 = 0.0219$, $\omega_1 = 0.0130$ | I vs. K | 9.44 | 0.002 |
| | L: the moth branch has $\omega_2 = 1$ | 14 | −6,121.45 | $\omega_2 = 1$, $\omega_1 = 0.0064$ | L vs. K | 1,174.6 | <0.001 |
| | M: each branch has its own ω | 23 | −5,525.41 | Variable ω by branch | I vs. M | 26.92 | 0.003 |
| | **Branch-site models** | | | | | | |
| | N: the moth branch | 16 | −5,513.53 | | | | |
| | O: the moth branch has ω = 1 | 15 | −5,513.53 | | N vs. O | 0 | 1 |

**Notes:**
[a] Number of parameters.
[b] The natural logarithm of the likelihood value.
[c] Twice the log likelihood difference between the two models.

was much higher than that of butterflies. This suggested that the moth *cry2* genes might have accumulated higher proportion of nonsynonymous mutations than the butterfly *cry2* in the evolutionary history, suggesting a less strong purification selection than *cry1*. However, no positive selection could be detected by PAML.

Then we calculated the *dn/ds* for all the transcript isoforms of the *cry1* and the *cry2* (as shown in Tables S3 and S4). Surprisingly, highly contrasting results were obtained. The *dn/ds* of *Hacry1* transcripts was 0.0082, which is in line with the expectation based on results of the *cry1*. Four replacement changes and 127 synonymous changes were detected. However, for the *cry2*, *dn/ds* increased dramatically to 0.684. Fourteen replacement changes and eight synonymous changes were found. It indicated that nonsynonymous mutations were accumulated with much higher rate in these *cry2* isoforms than in the *cry1* genes.

## Expressions *Hacrys* in different tissues, adult instars, populations and months

According to our preliminary experiment and experience investigating gene expression pattern in *H. armigera* (Mao et al., 2007; Sui et al., 2009; Zhang et al., 2011), β-actin was selected as reference gene in this study. Individuals of females and males from CD were used to detect tissue distribution of *Hacrys* mRNA transcripts including antenna, head, thorax, abdomen and leg, by quantitative RT-PCR analyses. Expressions were detected in all examined tissues but no statistically significant difference was detected among them (ANOVA: for *Hacry1*, in day time, $F(4,19) = 0.92$, $p = 0.4731$; in night time: $F(4,18) = 0.57$, $p = 0.6862$; for *Hacyr2*, in day time, $F(4,19) = 0.61$, $p = 0.6628$, in night $F(4,18) = 0.57$, $p = 0.6905$) (Figs. 3A and 3B).

Initially, the expression levels of both *Hacry1* and *Hacry2* were not significantly different in female or male adults. However, the expression level of *Hacry1* mRNA was significantly higher in male than female adults during days 3, 5, and 7 (Student's *t*-test, day 3, $t = 2.1696$, $p = 0.0365$; day 5, $t = 1.9515$, $p = 0.0494$; day 7, $t = 2.3900$, $p = 0.0270$). The expression level of *Hacry2* mRNA was significantly higher in males on day 1 (Student's *t*-test, $t = 2.1190$, $p = 0.0392$;) and day 5 (Student's *t*-test, $t = 3.6869$, $p = 0.0071$) than female adults (Figs. 3C and 3D), which might be due to the differences of rhythm and biological function between females and males. From CD female and male adults, the *Hacry1* and *Hacry2* mRNA level were significant higher than other populations including LF, 96s, 1,463 and XX (ANOVA: for *Hacry1*, $F(5,30) = 6.63$, $p = 0.0003$; for *Hacry2*, $F(5,26) = 2.69$, $p = 0.0436$) (Figs. 3E and 3F). The *Hacry1* genes in CD were significantly differently expressed among individuals from different months ($F(4,43) = 3.337$, $p = 0.0182$), however, there were no significant differences in the expression levels of *Hacry2* genes ($F(4,41) = 0.4738$, $p = 0.7545$) (Figs. 3G and 3H).

To investigate the effect of heredity, we detected the mRNA levels of *Hacrys* using CD third generation reared indoor (CDF3). The level of *Hacry1* mRNA was significantly down-regulated in CDF3 (Bonferroni multi comparison, $p = 0.009$ between CD and CDF3) and similar to other populations detected in this study (Figs. 3E and 3F). However, the levels of *Hacry2* mRNA were stable between CD and CDF3 (Bonferroni multi comparison,

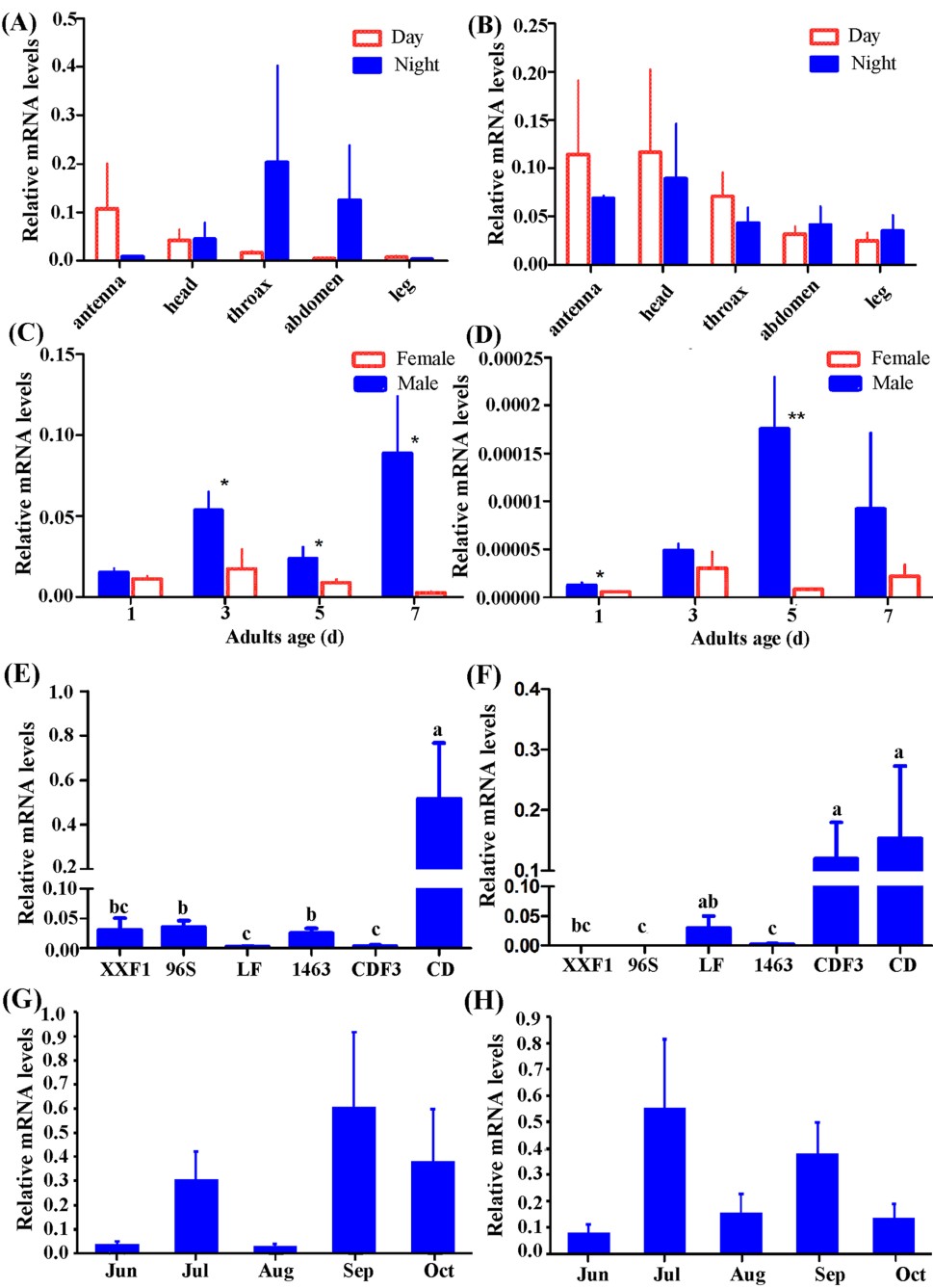

**Figure 3 Relative expression level of *Hacry1* and *Hacry2* in different tissues (*n* = 3–5), day instar stage (*n* = 4), population (*n* = 4–8) and months (*n* = 8–11) of adults.** (A) Relative expression level of *Hacry1* in different tissues of adults. (B) Relative expression level of *Hacry2* in different tissues of adults. (C) Relative expression level of *Hacry1* in different day instar stage of adults. (D) Relative expression level of *Hacry2* in different day instar stage of adults. (E) Relative expression level of *Hacry1* in different population of adults. (F) Relative expression level of *Hacry2* in different population of adults. (G) Relative expression level of *Hacry1* in different months of adults. (H) Relative expression level of *Hacry2* in different months of adults. Mean ± SE. The "*" and "**" denote statistical significance of the expression levels in (C) and (D) ($p < 0.05$ and $p < 0.01$, respectively). The different letters were used to show significant difference in (E) and (F) ($p < 0.05$, by Student's *t*-test).

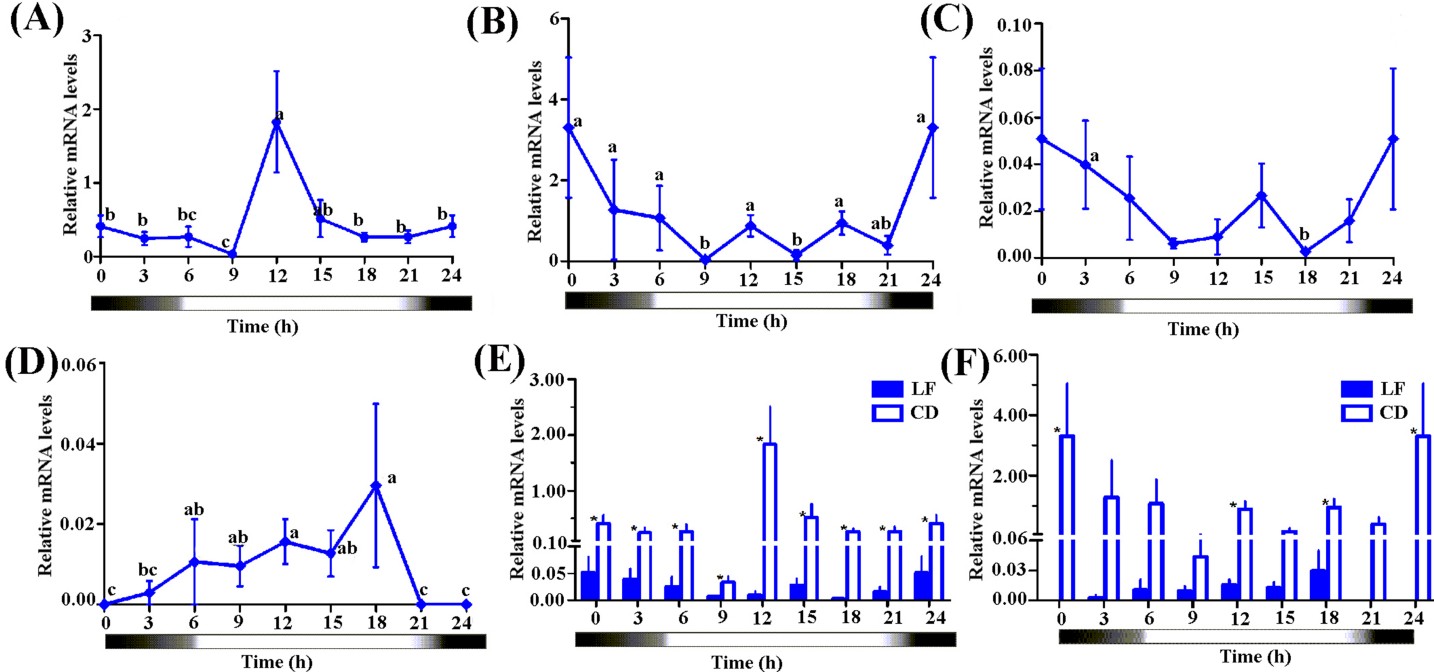

**Figure 4** **The expression level of *Hacrys* in CD and LF population.** (A) Diel changes of relative expression level of *Hacry1* in CD population. (B) Diel changes of relative expression level of *Hacry2* in CD population. (C) Diel changes of relative expression level of *Hacry1* in LF population with moths at 3-day after eclosion. The different letter indicates significant differences between groups (*p* < 0.05, by ANOVA). (D) Diel changes of relative expression level of *Hacry2* in LF population with moths at 3-day after eclosion. The different letter indicates significant differences between groups (*p* < 0.05, by ANOVA). (E) Relative expression level of *Hacry1* between CD and LF individuals. Mean ± SE. The "*" denote statistical significance of the expression levels between CD and LF individuals at the same time (*p* < 0.05, *n* = 3–4). (F) Relative expression level of *Hacry2* between CD and LF individuals. Mean ± SE. The "*" denote statistical significance of the expression levels between CD and LF individuals at the same time (*p* < 0.05, *n* = 3–4).

*p* = 0.686 between CD and CDF3), which were significantly higher than other populations except for LF (Figs. 3E and 3F).

## Diel changes in the expression of *cry1* and *cry2* from *H. armigera* in migrating and reared populations

We selected CD and an artificial reared population (LF, 3rd day moths post eclosion) to detect the diurnal fluctuation of the expression levels of *Hacry1* and *Hacry2*. qPCR analysis revealed a significant fluctuation of the two *crys* in both CD and LF. The two populations had a similar rhythm of *Hacry1* expression level from 0:00 to 12:00. However they show different profiles, with peaks and troughs at 12:00 and 9:00 for outdoor population (CD), 0:00 and 18:00 for indoor population (LF), respectively (Figs. 4A–4D).

The diurnal-rhythm of *Hacry2* mRNA was different between CD and LF, which was low during the day in CD while was high in LF at 0:00 and 18:00 (Figs. 4A–4D). The amount of *Hacry1* mRNA expression in CD was significantly higher than in LF in all of the detected samples except for at 6:00 and 15:00 (Figs. 4E and 4F). The amount of *Hacry2* mRNA did not show significant temporal variance in LF, but was significantly higher in CD at 0:00, 12:00 and 18:00 than in LF (Figs. 4E and 4F).

## DISCUSSION

Recently, *Yan et al. (2013)* reported two types of *cry* genes from *H. armigera*, each with one transcript respectively. Here, we identified six novel transcripts of *cry1* and seven novel transcripts of *cry2*. The transcripts of HaCRY1s contained a conserved PHR motif and the alternative splicing region located in the CCE domain which was also found in *cry1*s of *Mamestra brassicae* and *Spodotera exigua*, suggesting that the *cry1* genes might be evolving at a special period in Lepidoptera (*Merlin et al., 2006*; *Berndt et al., 2007*).

Similarly, the seven detected isoforms of the *Hacry2* are diverse in sequence length and nucleotides, which may be from alternative splicing of the same gene or different genes derived from duplication events. The CRY2s of insects act as a transcriptional repressor in a light-independent manner (*Zhu et al., 2005*; *Yuan et al., 2007*; *Ikeno, Numata & Goto, 2011*). The CCE domain is key to CRYs maintenance and can be active in the absence of a PHR domain (*Fankhauser & Ulm, 2011*; *Mazzotta et al., 2013*). Therefore, we suggest that HaCRY2-1, HaCRY2-2, HaCRY2-3 and HaCRY2-4 might be function like other HaCRY2s, despite lacking a complete PHR domain.

Transcription analyses of *cry1* and *cry2* from insects including *Drosophila melanogasta*, *Apis mellifera*, *Riptortus pedestris*, *M. brassicae*, *Danaus plexippus* revealed that each insect species contained only one copy of *cry1* and/or *cry2* without alternative splicing (*Emery et al., 1998*; *Zhu et al., 2005*; *Merlin et al., 2006*; *Rubin et al., 2006*; *Ikeno, Numata & Goto, 2008*). The *cry* genes of *H. armigera* were the most polymorphic among all of the known insect *cry* genes, which might be related to the lifestyle and environmental adaptability of *H. armigera*. *dn/ds* analysis did not detect any positive selective evolution between the *cry* genes of nocturnal moths and diurnal butterflies. However, the *dn/ds* analysis of *H. armigera* transcripts suggested that the encoding sequence of the *Hacry1* was negatively selected possibly due to functional constraint, but the *cry2* showed less strong selection than *cry1*. For the branch-site model, the neutral model could not be rejected. These results suggested that the *Hacry1* might play a more important role than the *Hacry2* in *H. armigera*.

mRNA levels of *Hacrys* were neither tissue-specific nor developmental-stage-specific (*Yan et al., 2013*). Based on ovary development, most of the individuals from CD population were 3–4 days instars after eclosion. Therefore, we used whole individuals of 3 days adult instar in laboratory populations to compare with wild CD individuals. To our surprise, the mRNA levels of the *cry1* were significantly higher in CD than LF while the *Hacry2* was not as strong as the *Hacry1*.

To confirm that the levels of *Hacrys* transcripts were correlated to the migrating behavior, we reared offspring of CD for three generations (CDF3) in laboratory and investigated the mRNA levels of *Hacrys* using the 3rd day adults after eclosion. The expression level of *Hacry1* was significantly down-regulated in CDF3 population, suggesting that HaCRY1 might play a role in the migrating behavior of *H. armigera*. Detected expression of *Hacry2*, however, was not significantly changed in CDF3, suggesting that the *cry2* might not have specific function in the migrating behavior of *H. armigera*. Recently, CRY proteins were reported as magnetoreceptors and using light signal for their

function in navigation (*Baik et al., 2017*; *Marley et al., 2014*; *Bazalova et al., 2016*; *Rodgers & Hore, 2009*; *Gegear et al., 2010*). Our results warrant further investigation on the functions of CRY proteins in the migrating behavior of moths, in areas of both circadian biology and magnetochemical biology. This potential role in migration may have important implications for our understanding of migratory behavior of key crop pest species.

## CONCLUSIONS

We found the *Hacry1* and *Hacry2* of *H. armigera* occurred alternative splicing which generated six and seven transcripts, respectively. Selection pressure analysis suggested that the *Hacry1* was under purifying selection by a strong negative selection pressure whilst the *Hacry2* was less constrained and showed a less strong purification selection than *cry1*. At mRNA levels by qPCR, *Hacrys* were more abundantly transcribed in wild migrating populations than in laboratory maintained populations, and expression of the *Hacry2* was lower than *Hacry1* in all samples tested. Interestingly, when compared with the migrating parental population, offspring reared in laboratory conditions showed a significant reduction on transcription of the *Hacry1* but not *Hacry2*. These results suggest that *Hacry1* is more related to the migration of *H. armigera* than *Hacry2*.

## ACKNOWLEDGEMENTS

We would like to thank Dr. Xiaowei Fu for collecting samples on Beihuang Island.

### Funding

This work was supported by the Agricultural Science and Technology Innovation Program (ASTIP-TRIC04), the Key S&T project of China National Tobacco Corporation (110201601022(LS-02)), and the Science Foundation for Young Scholars of Institute of Tobacco Research of CAAS (2015B03). The funders had no role in study design, data collection and analysis, decision to publish, or preparation of the manuscript.

### Grant Disclosures

The following grant information was disclosed by the authors:
Agricultural Science and Technology Innovation Program: ASTIP-TRIC04.
Key S&T project of China National Tobacco Corporation: 110201601022(LS-02).
Science Foundation for Young Scholars of Institute of Tobacco Research of CAAS: 2015B03.

### Competing Interests

The authors declare that they have no competing interests.

### Author Contributions

- Liyu Yang conceived and designed the experiments, performed the experiments, prepared figures and/or tables, authored or reviewed drafts of the paper, approved the final draft.

- Yingjie Liu conceived and designed the experiments, performed the experiments, prepared figures and/or tables, authored or reviewed drafts of the paper, approved the final draft.
- Philip Donkersley analyzed the data, authored or reviewed drafts of the paper, approved the final draft.
- Pengjun Xu conceived and designed the experiments, performed the experiments, analyzed the data, prepared figures and/or tables, authored or reviewed drafts of the paper, approved the final draft.

## Data Availability

The sequence data is available at NCBI: JN997418, JQ713131, JQ713132, JQ713133, JQ713134, JQ713135, JQ713136, JQ713137, JQ713138, JQ713139, JQ713140, JQ713141, JQ713142.

## Supplemental Information

Supplemental information for this article can be found online at http://dx.doi.org/10.7717/peerj.8071#supplemental-information.

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
