# Peer review of "Up-regulation of cryptochrome 1 gene expression in cotton bollworm (Helicoverpa armigera) during migration over the Bohai Sea"

_PeerJ, doi:10.7717/peerj.8071_

## Round 0.1 · original submission · Major Revisions

Dear Dr. Yang and colleagues:

Thanks for submitting your manuscript to PeerJ. I have now received three independent reviews of your work, and as you will see, the reviewers raised some concerns about the research. Despite this, these reviewers are optimistic about your work and the potential impact it will lend to research on migration-induced cryptochrome 1 gene expression in cotton bollworm. Thus, I encourage you to revise your manuscript, accordingly, taking into account all of the concerns raised by the reviewers.

While the concerns of the reviewers are relatively minor, this is a major revision to ensure that the original reviewers have a chance to evaluate your responses to their concerns.

In your revision, please provide more information on the sampling regions and their association with moth migratory routes. Also, provide more evidence for the conclusions that you reach. Avoid statements that are not strongly supported by your data.

I look forward to seeing your revision, and thanks again for submitting your work to PeerJ.

Good luck with your revision,

-joe

Reviewer 1 ·

Basic reporting

The English is clear, Introducion to show context and Literature well referenced and relevant. Structure conforms to PeerJ standards. Figures are relevant, high quality, well labelled and described.

Experimental design

Methods described with sufficient detail and information to replicate.

Validity of the findings

Conclusions is ok.

Additional comments

The research paper entitled, “Up-regulation of cryptochrome 1 gene expression in cotton bollworm (Helicoverpa armigera) 
during migrating over the Bohai Sea” is every helpful to design the migrating behavior in H. armigera. The objectives are nicely framed with defined goal.

Reviewer 2 ·

Basic reporting

Yang et al. reported the finding of multiple isoforms of cry1 and cry2 in H. armigera. Both genes are more expressed in wild migratory populations than those maintained in lab. cry2 is in general less constrained and less expressed than cry1.

There are a few major problems with figure 3 and 4. The numbering is flipped, and some subfigures are not covered in the main text. More details should be provided either in caption or in the methods part. Besides, high-resolution figures are preferred. Font and size of the labels should be adjusted to the same format among panels. Grammar errors in the captions should be corrected before acceptance.

Experimental design

More details of the samples should be added (see General comments for the author).

Validity of the findings

The findings are clear, however, certain implications are not well supported. Authors should either provide more evidence to support those conclusions or cut/ move them to discussion.

Additional comments

Major changes:

1. In general, the authors should be more conservative on drawing conclusions.
a. More evidence is needed to support that "relaxation on nonsynonymous constraint suggested a possibility that the cry2 might have lost some function in H. armigera" (Line 188, Line 235). The authors could either point out nonsense mutations on cry2 or drop this statement.
Moreover, this loss-of-function statement contradicts with the discussion that HaCRY2s may function the same as other HaCRYs (Line 223-224).
b. It's unconvincing that cry2 has undergone a neutral selection given the small omega. It shows less strong purification selection than cry1, but not to the degree to be considered as neutral.
c. There are a number of factors other than migration that could potentially down regulated cry1 and cry2 during the domestication in lab, e.g. the luminosity and frequency of the light that was used, the stableness of the light cycle, the temperature and humidity, etc. There is not enough evidence supporting that cry1 plays an important role in migration given that migration is not the only difference between the wild and lab populations. This part is suggested to be moved to discussion from results.


2. Questions concerning the phylogenetic analysis:
a. What will happen if the authors include all transcripts in the phylogenetic analyses? This could potentially help resolve the inconsistence of dn/ds between the analyses with and without all isoforms. At least the authors should provide more details like what other outgrip species are included in the analysis with all transcripts.
b. The author mentioned separation the nucleotide sequences to three partitions, but could the author clarify which partition (1st/ 2nd/ 3rd codon) are the phylogenetic results based on?
c. Is it possible to use at least one common outgroup species for both cry1 and cry2? (e.g. mosquitos, butterflies, or moths)


3. When the authors refer to Figure 3a they might mean Figure 4. The numbering needs to be corrected.

4. There are a few major problems concerning figure 3. It's strongly suggested that the authors double check the figures to clearly convey the right information.
a. Figure 3b and 3d are not mentioned in the main text.
b. Assuming the y axis represents the relative expression of cry1/2 to beta actin, Figure 3a doesn't show the expression difference between CD and LF. If the y axis represents the relative expression of cry1/2 of LF and CD, it should be noted in both the axis label, figure caption, and main text. If the authors actually mean Figure 4, what should be the correct text for figure 3a?
c. Figure 3b shows difference between males and females. Could the author possibly explain the huge difference?
d. Among the four panels, the expression of cry1 is sort of stable < 0.2, but the relative levels of cry2 varies too much, from 0.0001 to 0.8. Could the author explain the difference?
e. Given that the expression of cry1 and cry2 varies a lot in different sexes (3b), times of the day (3a) and year (3d), days of development (3b), tissues (3a), the authors should provide more details about controlling of the variables, for example, which sex and tissue were used for figure 3a and 3c, etc.? This question also applies to Figure 4.


Minor changes:
1. Line 15, Line 31, It's not necessary to mention flavoprotein here since it's not relevant to the main findings of this study.
2. Line 19, please first define the abbreviation Hacry before using it.
3. Line 38-40, This piece of information is not closely relevant and should either be modified or removed.
4. Line 41-46, could the authors further explain if there is any difference in the regulation of circadian rhythm among the three categories?
5. Line 54, remove "within China".
6. Line 58, overwinter is one word.
7. Line 97, "Plasmids of Hacry positive clones" --> "positive clones of plasmids of Hacry".
8. Line 116, adapt to.
9. Line 257, "... H. armigera occurred alternative splicing generated six and seven transcripts respectively." --> "occurred alternative splicing which generated six and seven transcripts, respectively."
10. Line 259, constraint --> constrained
11. Line 306, Red shown the PHP region --> region shaded in red correspond to the PHP region
12. Line 307, "--/./- stand for ..." --> "--/./- stands for ..."
13. Line 308, I don't see == in the figure?
14. Line 390, larvae should not be italicized.
15. The axes labels of 4b Cry1 CD are smaller than the rest.
16. Could the authors provide a high-resolution Figure S1?
17. Is there a better way to annotate MTHF and FAD region? The current underscores is hard to tell.
18. In a few figures, panel table (A)(B) have different fonts and sizes from the figure. It's recommended to adopt one format.

·

Basic reporting

The overall writing really needs a little improvement. It would be hard for people whose native language is not English. At least information should be delivered accurately to readers. Please spend time to polish it on your grammar and sentence structure.

Technical concerns:
1. Page 1 line 17, “identify” should be “identified”, and there are some similar mistakes throughout the rest of this article, please correct them.
2. Page 2 line 23, “than in laboratory maintained populations”, “that” should be inserted followed by “than”, and there are some similar mistakes throughout the rest of this article, please correct them.
3. Page 3 line 31, I think there is something wrong with the first sentence.
4. Page 3 line 39-40, there is something wrong with the sentence, please express it correctly and clearly.
5. Page 3 line 43-45, there is something wrong with the sentence, please express it correctly and clearly.
6. Page 4 line 57, “over wintering” should be “overwintering”, it would be better to keep consistent with preceding part of the manuscript.
7. Page 4 line 64, there is something wrong with the sentence, please express it correctly and clearly.
8. Page 5 line 72, there is something wrong with the sentence, please express it correctly and clearly.
9. Page 5 line 76, please change “25℃” to “25 ℃”. And there are some similar mistakes throughout the rest of this article, please correct them.
10. Page 5 line 82, please change “Hacry” to “Hacry”.
11. Page 6 line 111-112, there is something wrong with the sentence, please express it correctly and clearly. And there are some similar mistakes throughout the rest of this article, please correct them.
12. Page 7 line 132, I think that “by us” is redundant.
13. Page 8 line 148, I think that “by us” is redundant. there is something wrong with the sentence, please express it correctly and clearly.
14. Page 9 line 167, “Phologenetic” should be “Phylogenetic”. Please find similar mistakes and correct them.
15. Page 11 line 199-200, It is confusing. Please describe it correctly and clearly.
16. Page 11 line 213-214, there is something wrong with the sentence, please express it correctly and clearly.
17. Page 12 line 230, there is something wrong with the sentence, please express it correctly and clearly.
18. Page 12 line 235, there is something wrong with the sentence structure. please express it correctly and clearly.
19. Page 13 line 244-245, please correct their tenses. Please find similar mistakes and correct them.
20. Letter case in figures and figure legends should be same. Statistical analysis should be consistent and conform to standards. And “p” should be lower case and Italic.
21. Page 14 line 279, gene name should be Italics, please correct them and the others of this article.
22. In some figures, it is better to add title in vertical axis and in each of bar graph.

Experimental design

My major concern with this paper is that the title is “migrating over Bohai sea”, but the readers would not find Bohai sea throughout the manuscript. International readers will be hard to understand the sampling regions and their association with migrating. And this should be put into the introduction, methods, results, and discussion.

Validity of the findings

Cryptochromes are flavoproteins and play a pivotal role in circadian clocks which mediate behavior of organisms such as feeding, mating, and migrating navigation. In this manuscript, the authors identified novel transcripts in H. armigera by sequencing, six isoforms of cry1 and seven isoforms of cry2. Phylogenetic analysis showed that the transcripts of cry1 and cry2 align closely with other insect homologs, indicating within-species divergence of Hacry. A dn/ds analysis revealed that the encoding sequence of the cry1 was under purifying selection by a strong negative selection pressure whereas the cry2 was less constraint and underwent a nearly neutral evolution. Hacrys were more abundantly transcribed in wild migrating populations than that in laboratory maintained populations, and expression of the cry2 was lower than cry1. Moreover, when compared with the migrating parental population, offspring reared in laboratory conditions showed a significant reduction on transcription of the cry1 but not cry2. Taken together, they concluded that cry1 plays a more important role than cry2 during the migration behavior of H. armigera, which may have important implication for understanding migratory behavior of key crop pest species.

Additional comments

This manuscipt needs revision before acceptance.

---

## Round 0.2 · accepted · Accept

Dear Dr. Yang and colleagues:

Thanks for re-submitting your revised manuscript to PeerJ, and for addressing the concerns raised by the reviewers. I now believe that your manuscript is suitable for publication. Congratulations! I look forward to seeing this work in print, and I anticipate it being an important resource for research on migration-induced cryptochrome 1 gene expression in cotton bollworm.

Thanks again for choosing PeerJ to publish such important work.

-joe

Reviewer 2 ·

Basic reporting

The authors have properly responded to my comments and nicely corrected grammatical errors. I don't have other concerns at the moments.

Experimental design

As above.

Validity of the findings

As above.

Additional comments

As above.

·

Basic reporting

The authors have answered all concerns raised in the last review.

Experimental design

no comment

Validity of the findings

no comment

Additional comments

The authors have answered all concerns, and I suppor the publication of this manuscript.